# The physical dimensions of amyloid aggregates control their infective potential as prion particles

**Ricardo Marchante, David M Beal, Nadejda Koloteva-Levine, Tracey J Purton, Mick F Tuite, Wei-Feng Xue\***

Kent Fungal Group, School of Biosciences, University of Kent, Canterbury, United Kingdom

**Abstract** Transmissible amyloid particles called prions are associated with infectious prion diseases in mammals and inherited phenotypes in yeast. All amyloid aggregates can give rise to potentially infectious seeds that accelerate their growth. Why some amyloid seeds are highly infectious prion particles while others are less infectious or even inert, is currently not understood. To address this question, we analyzed the suprastructure and dimensions of synthetic amyloid fibrils assembled from the yeast (*Saccharomyces cerevisiae*) prion protein Sup35NM. We then quantified the ability of these particles to induce the [$PSI^+$] prion phenotype in cells. Our results show a striking relationship between the length distribution of the amyloid fibrils and their ability to induce the heritable [$PSI^+$] prion phenotype. Using a simple particle size threshold model to describe transfection activity, we explain how dimensions of amyloid fibrils are able to modulate their infectious potential as prions.

DOI: https://doi.org/10.7554/eLife.27109.001

## Introduction

The formation, processing, deposition and propagation of amyloid are at the center of a number of disease-associated and functional biological phenomena. While some amyloid aggregates are associated with devastating neurodegenerative diseases such as Alzheimer's, Parkinson's and prion diseases (*Eisenberg and Jucker, 2012*; *Knowles et al., 2014*), other amyloid structures play many functional biological roles such as bacterial biofilm formation (*Larsen et al., 2007*; *Romero et al., 2010*; *Chapman et al., 2002*), mammalian melanosome organization (*Berson et al., 2003*) or epigenetic inheritance in yeast (*Tuite and Serio, 2010*). Some amyloid aggregates are recognized as prions because they form infectious particles, seeds that have the ability to be transmitted between cells, organs, individuals or even species (*Aguzzi et al., 2007*; *Prusiner, 1991*; *Sabate et al., 2015*). Prions can subsequently propagate by converting their soluble and correctly folded protein counterparts into the amyloid fold (*Prusiner, 1991*). However, this conversion process is not exclusive to prions as amyloid growth by elongation at fibril ends of preformed seeds recruits and converts soluble protein counterparts independently of the fibrillar seeds being transmissible prion particles or non-infectious amyloid.

The molecular and cellular mechanisms underlying prion infectivity are not fully understood, but have been a key focus of research, particularly for the mammalian prion protein (PrP) which can be transmitted between hosts across different species (*Aguzzi et al., 2007*; *Weissmann et al., 2011*). Indeed, there is compelling evidence that the transmission of PrP amyloid between cattle with Bovine Spongiform Encephalopathy and humans gave rise to a number of cases of so-called variant Creutzfeldt-Jakob disease (*Collinge, 1999*). Furthermore, recent studies have shown that several disease-associated amyloid aggregates such as α-synuclein (Parkinson's Disease), Tau (Alzheimer's

**\*For correspondence:**
w.f.xue@kent.ac.uk

**Competing interests:** The authors declare that no competing interests exist.

disease and other tauopathies), amyloid-β (Alzheimer's disease) and huntingtin (Huntington's disease) are also capable of cross-cell transmission in a 'prion-like' manner (reviewed in [*Aguzzi and Lakkaraju, 2016*]). The possibility that such aggregates might also be infectious under some circumstances has also been suggested (*Jaunmuktane et al., 2015*). These new data highlight a gap in the knowledge of the factors that govern the infectious potential of amyloid particles in general. Why are some amyloid fibrils highly infectious prion particles while others are less infectious or even inert?

Amyloid fibrils are stable supramolecular structures characterized by the cross-beta molecular architecture supported by intermolecular hydrogen bonds parallel to the fibril axis (*Eisenberg and Jucker, 2012*; *Knowles et al., 2014*). Their formation is usually preceded by protein misfolding events that result in the emergence of assembly-competent forms of amyloid-forming proteins (*Figure 1*). Amyloid formation then proceeds through nucleated polymerisation and fibril elongation with growth occurring by templated addition of monomeric protein to the fibril ends (*Serio et al., 2000*; *Collins et al., 2004*). The fragmentation of amyloid fibrils is an important secondary process that divides fibrils and increases the number of growth competent fibril ends, which consequently will accelerate amyloid growth (*Xue et al., 2008*; *Knowles et al., 2009*). This secondary process can also increase the cytotoxic potential of amyloid particles (*Xue et al., 2009a*), as several studies have

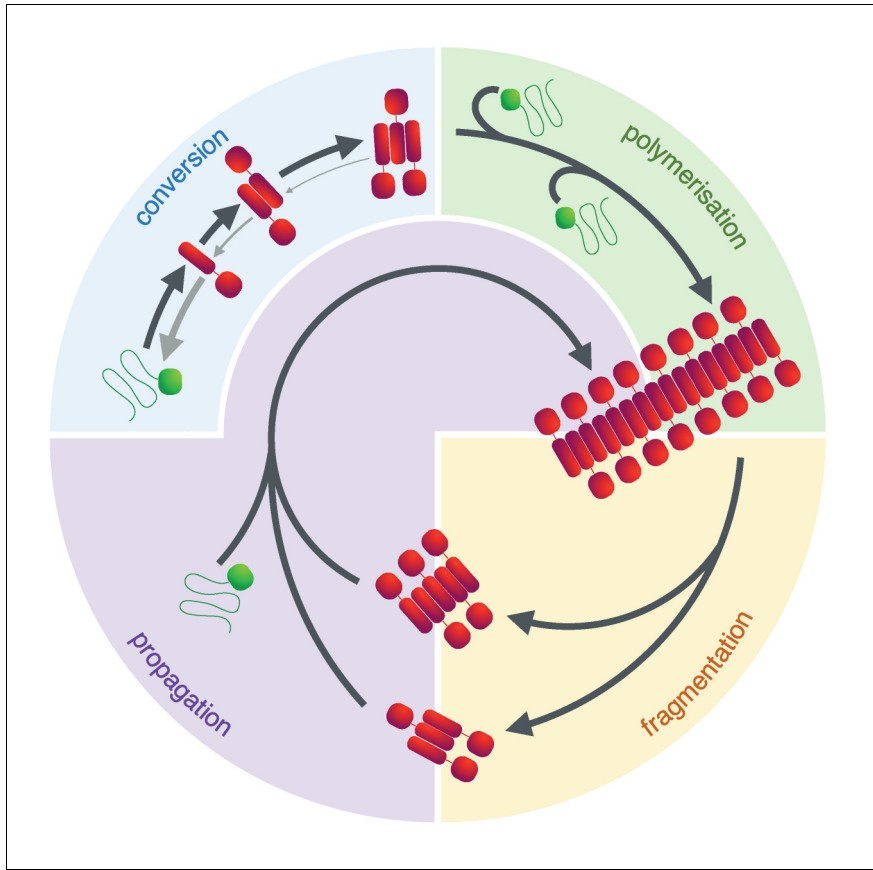

**Figure 1.** The prion lifecycle. The de novo formation of a prion involves reversible protein misfolding and consequential formation of amyloid conformers, which can assemble into small oligomeric species. Once this assembly reaches a critical size (i.e. the nucleus size), further polymerisation is favoured and fibrils grow by addition of protein monomer to their ends. This process can be accelerated by fibril fragmentation which increases the number of fibril ends to which monomer can be added. In the yeast *Saccharomyces cerevisiae* this is accomplished by the molecular chaperone Hsp104 and its co-chaperones and is essential to ensure a suitable number of prion particles (propagons) of the appropriate size is generated, ensuring their transmission during cell division. The process of fragmenting and transmitting prion particles is usually referred to as propagation.
DOI: https://doi.org/10.7554/eLife.27109.002

correlated smaller amyloid species with higher amyloid toxicity (*Tipping et al., 2015*). Despite its roles in disease-associated amyloid systems, fibril fragmentation is also crucial for ensuring the maintenance of functional amyloid such as yeast prions by producing sufficient number of growth competent amyloid particles to propagate the amyloid conformation and the prion phenotype (*Tanaka et al., 2006*; *Wang et al., 2011*).

The budding yeast *Saccharomyces cerevisiae* encodes a large number of functional prion-forming proteins, one of which is the translation termination factor Sup35 that, in its prion form, gives rise to the associated prion phenotype [*PSI*$^+$] (*Tuite and Serio, 2010*; *Wickner, 1994*; *Patino et al., 1996*). Translation termination occurs efficiently in prion-free cells ([*psi*$^-$]), but is impaired when Sup35 switches to its prion conformation (*Wickner, 1994*; *Patino et al., 1996*; *Cox, 1965*; *Glover et al., 1997*), making the Sup35 prion protein an epigenetic regulator of gene expression. In this system, the action of molecular chaperones, including Hsp104, catalyzes the fragmentation of the amyloid fibrils and ensures that sufficient amyloid particles (referred to as propagons) exist during cell division so that the prion phenotype is propagated to subsequent generations (*Chernoff et al., 1995*; *Glover and Lindquist, 1998*) (*Figure 1*). Crucially, fully synthetic prion particles formed in vitro from a recombinant form of Sup35 encompassing the first 253 residues of the protein (designated Sup35NM) are capable of being transfected into yeast cells and inducing the [*PSI*$^+$] prion phenotype (*King and Diaz-Avalos, 2004*; *Tanaka et al., 2004*). This has provided us a highly tractable system that has allowed us to carry out a quantitative biophysical assessment of the factors that govern the infective potential of amyloid and prion particles.

Here, employing the Sup35NM/[*PSI*$^+$] yeast prion system, we report a quantitative investigation into the infective potential of prion particles. We ask whether the dimensions and physical properties of prion particles can modulate their infective potential. This is achieved by quantifying the in vivo response of yeast cells to synthetic Sup35NM prion particles formed in vitro with recombinant Sup35NM and which have tailored length distributions by controlled fibril fragmentation. We use quantitative atomic force microscopy (AFM) and single-particle image analysis to characterize the morphology and suprastructure of the synthetic Sup35NM prion particles, and resolve their size distribution and particle concentration to a high quantitative detail. We then quantify the potential of these synthetic prion particles to transfect yeast cells and induce the heritable [*PSI*$^+$] state in vivo. Our data show a striking relationship between the size distribution of prion particles and their ability to confer the prion phenotype associated with these particles. Detailed analysis of this relationship reveals that the ability to transfect and induce the [*PSI*$^+$] phenotype is not identical for individual Sup35NM prion particles of different lengths, and has allowed us to estimate the particle size threshold for Sup35NM transfection of yeast cells using a simple modelling approach. Our results indicate that the physical dimensions of otherwise identical prion particles are key parameters that are sufficient to alter the infective potential of prion particles. These conclusions suggest possible routes to reduce prion and prion-like infectivity of amyloid by aggregate size modification approaches, for example by promoting the formation of large inert aggregates from transmissible particles or by increasing the stability of fibrils to reduce the creation of transmissible particles by fibril fragmentation, in strategies to combat the transmission of prions and amyloid particles.

## Results

### In vitro formation and AFM imaging of Sup35NM amyloid fibrils

In order to analyze the effects of fibril dimensions on prion infectivity we first produced synthetic Sup35NM prion particles in vitro from recombinant monomeric Sup35NM, and generated fibril samples with a range of length distributions. The yeast prion protein Sup35 can be subdivided into three regions: N, M and C. The N (residues 1–123) and M region (124–253) are responsible for amyloid formation and prion maintenance, while the C terminal region of the protein (residues 254–685) is responsible for its translation termination function. When expressed in *E. coli*, the N + M regions (Sup35NM, residues 1–253) are sufficient to confer the [*PSI*$^+$] phenotype when transfected into [*psi*$^-$] yeast cells (*King and Diaz-Avalos, 2004*; *Tanaka et al., 2004*; *Sparrer et al., 2000*). Recombinant Sup35NM protein monomers were therefore expressed in *E. coli* and purified under denaturing conditions as described in Materials and methods. Sup35NM polymerisation reactions were carried out in 20 mM sodium phosphate buffer, pH 7.4, 50 mM NaCl at a protein concentration of 10 µM to

form Sup35NM amyloid fibril samples at a temperature of 30°C, which is the standard laboratory growth temperature for *S. cerevisiae*. Both de novo formation and polymerisation of Sup35NM under these conditions were monitored in parallel in reactions containing the fluorescent amyloid binding dye Thioflavin T (*Figure 2a*).

Under the conditions employed, the Sup35NM polymerization reaction progress curves showed a sigmoidal shape expected for amyloid formation, with a lag phase of approximately 5 hr, followed by an exponential growth phase of approximately 5–10 hr in length. The reactions reached the upper plateau phase after approximately 20–30 hr. Analysis of the resulting amyloid fibrils using AFM imaging after the reactions reached the upper plateau (*Figure 2b*, *upper left image*) showed suprastructures consisting of large, intertwined networks of long fibrils. We next fragmented these fibrils by controlled sonication (see Materials and methods, *Figure 2b*). After 5 s of mechanical perturbation by sonication, a number of shorter and more disperse fibrils and small fibril clusters compared with non-fragmented initial samples were observed by AFM. An increasingly dispersed and non-clustered fibril population was observed with further sonication. A range of sonication durations were tested to generate a range of fibril sizes confirmed by AFM imaging (*Figure 2b*).

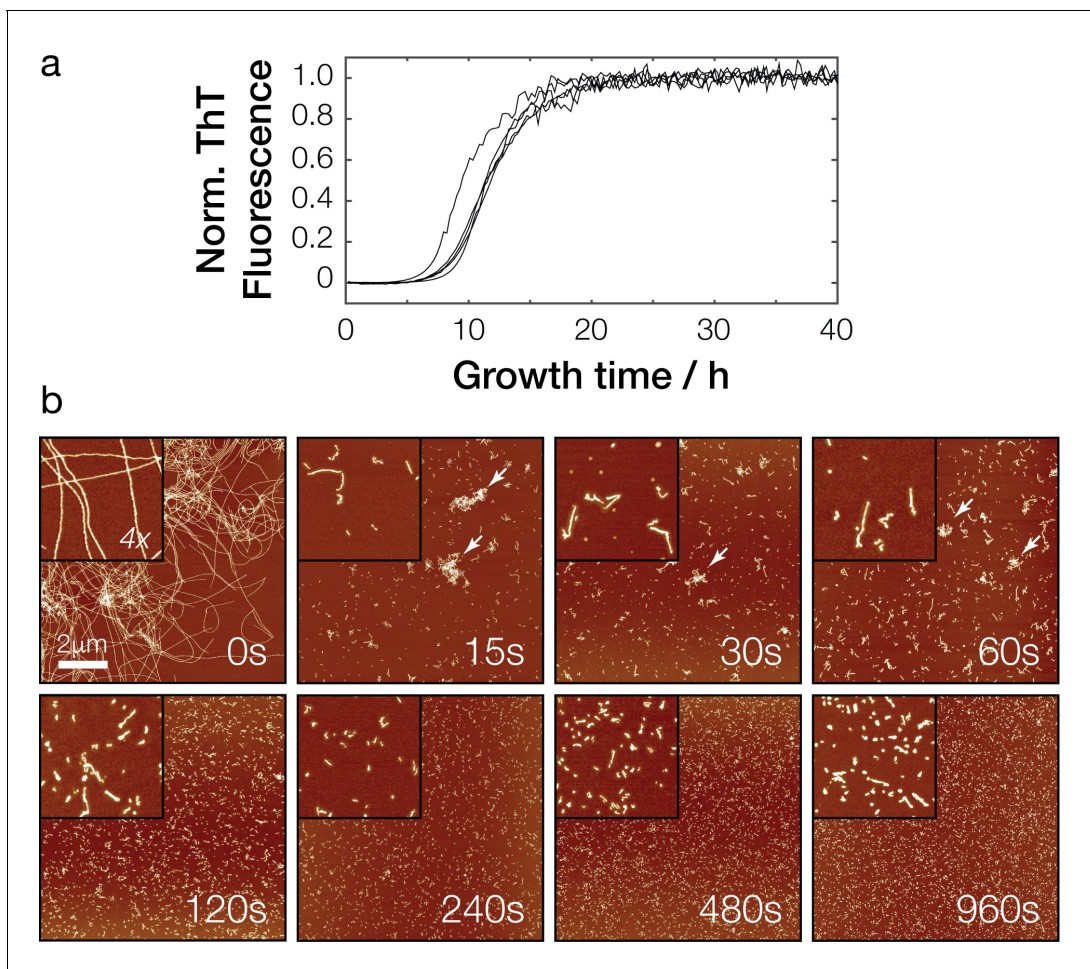

**Figure 2.** In vitro polymerization and fragmentation of Sup35NM prion fibrils. (a) Sup35NM polymerization monitored using the amyloid-binding dye Thioflavin T. Five experimental replicates are plotted, with curves normalized to their upper baseline. (b) Representative atomic force microscopy images of Sup35NM amyloid fibrils before (0 s) and after sonication. Samples were diluted 1:300 prior to deposition on the mica surface except for the 0 s sample. Images of 10 μm x 10 μm in scan size are shown together with 4 x further magnified views. The scale bar represents the length of 2 μm in all images and arrows show examples of clusters of fibril particles.
DOI: https://doi.org/10.7554/eLife.27109.003

## Characterization of Sup35NM prion particles

We next quantified the size distribution of the Sup35NM prion particles using a combination of sucrose density gradient analysis and semi-denaturing detergent agarose gel electrophoresis (SDD-AGE) (*Kryndushkin et al., 2003*). These biochemical methods have been previously used to distinguish prion aggregates in cell populations that have the prion phenotype versus those that do not, as well as to assess the occurrence of different prion conformational variants. Native sucrose density gradient analysis of Sup35NM amyloid fibrils fragmented to different extents by controlled sonication (*Figure 3a*) showed a clear shift in relative aggregate size after sonication. As seen in *Figure 3a*, fraction one containing monomeric Sup35NM was composed of less than 5% of total protein content in all samples, indicating that the polymerisation reaction had reached near-completion, and the controlled sonication had not caused increased free monomer concentration due to depolymerisation, as seen previously in other amyloid-forming systems (*Xue and Radford, 2013*). The bulk of the fibril material shifted from the heavier to lighter fractions when sonication time was increased, indicating a reduction in the size distribution of the prion particles.

The differences in size distribution post-sonication were more difficult to detect when the fibril samples were characterized by SDD-AGE (*Figure 3b*). SDD-AGE analysis is performed under semi-denaturing conditions that do not disrupt the amyloid core of the fibril particles. Thus, the SDD-AGE data, which showed the position of the core Sup35NM aggregates as broad bands, did not reveal any change in electrophoretic mobility with increasing sonication duration after the first 15 s of sonication. This may reflect the breakup of the large fibril network from the first 15 s of sonication as seen in *Figure 2b*. However, the data demonstrate that SDD-AGE cannot distinguish further shortening of the amyloid particles or breakup of small fibril clusters (*Figures 2b* and *3a*) after the initial break-up of the fibril network. Importantly, the SDD-AGE analysis (*Figure 3b*) did indicate that the relative aggregate size in the [*PSI*⁺] cell extract used as a control only overlap with the in vitro generated sample that was sonicated for more than 15 s, indicating that the in vivo prion particles do not resemble the fibril network seen in the untreated synthetic sample (*Figure 2b*, 0 s). This result is corroborated by the fact that the in vitro generated sample that was not treated (*Figure 2b*, 0 s) cannot induce the [*PSI*⁺] phenotype, while the [*PSI*⁺] cell extract typically is able to induce 10–40% [*PSI*⁺] cells when transfected into [*psi*⁻] yeast cells under the standard conditions for transfection employed (Materials and methods, *Figure 3—figure supplement 1*). Interestingly, the in vivo formed prion particles from the [*PSI*⁺] cell extract did overlap with the lower portions of the samples sonicated for longer periods than 15 s on SDD-AGE, suggesting that the prion species present in vivo may be represented by the lower molecular weight species in the in vitro assembled and sonicated fibril samples. The efficiency of a [*PSI*⁺] cell extract to induce [*PSI*⁺] cells when transfected into [*psi*⁻] yeast cells is highly variable due to many factors such as differences in protein expression levels of Sup35, the prion strain variant present as well as the variable repertoire of chaperones and other cellular factors that associate with the fibrils. Therefore, to isolate the effect of particle dimensions on their ability to transfect cells, we next analyzed in detail the dimensions of the in vitro assembled and sonicated fibril samples and assessed their ability to induce the [*PSI*⁺] state in [*psi*⁻] cells.

## Size distribution and particle concentration quantification of Sup35NM prion samples

To resolve the detailed length distribution and the size change of the amyloid particles in our samples in absolute length units, we next imaged and measured the length of individual particles using AFM and carried out length distribution analysis (*Xue and Radford, 2013*) on the Sup35NM fibril samples. This analysis was then used to estimate particle number concentration, particle widths (particle heights in AFM image data) and particle lengths for each individual synthetic prion sample. A total of 72 images were collected and analyzed for 26 samples, with at least two images and a minimum of 500 particles per sample analyzed to ensure sufficient numbers of fibril particles were taken into account (see *Table 1*).

In general, increased time of controlled mechanical perturbation by sonication resulted in a decrease of the mean particle size as previously shown (*Xue and Radford, 2013*; *Xue et al., 2009b*), with longer sonication times producing smaller mean fibril lengths (*Figure 4* and *Figure 4—figure supplement 1*). This can also be seen in the sample particle length distributions (*Figure 4a* and *Figure 4—figure supplement 1*), which show the presence of an increasing number of smaller

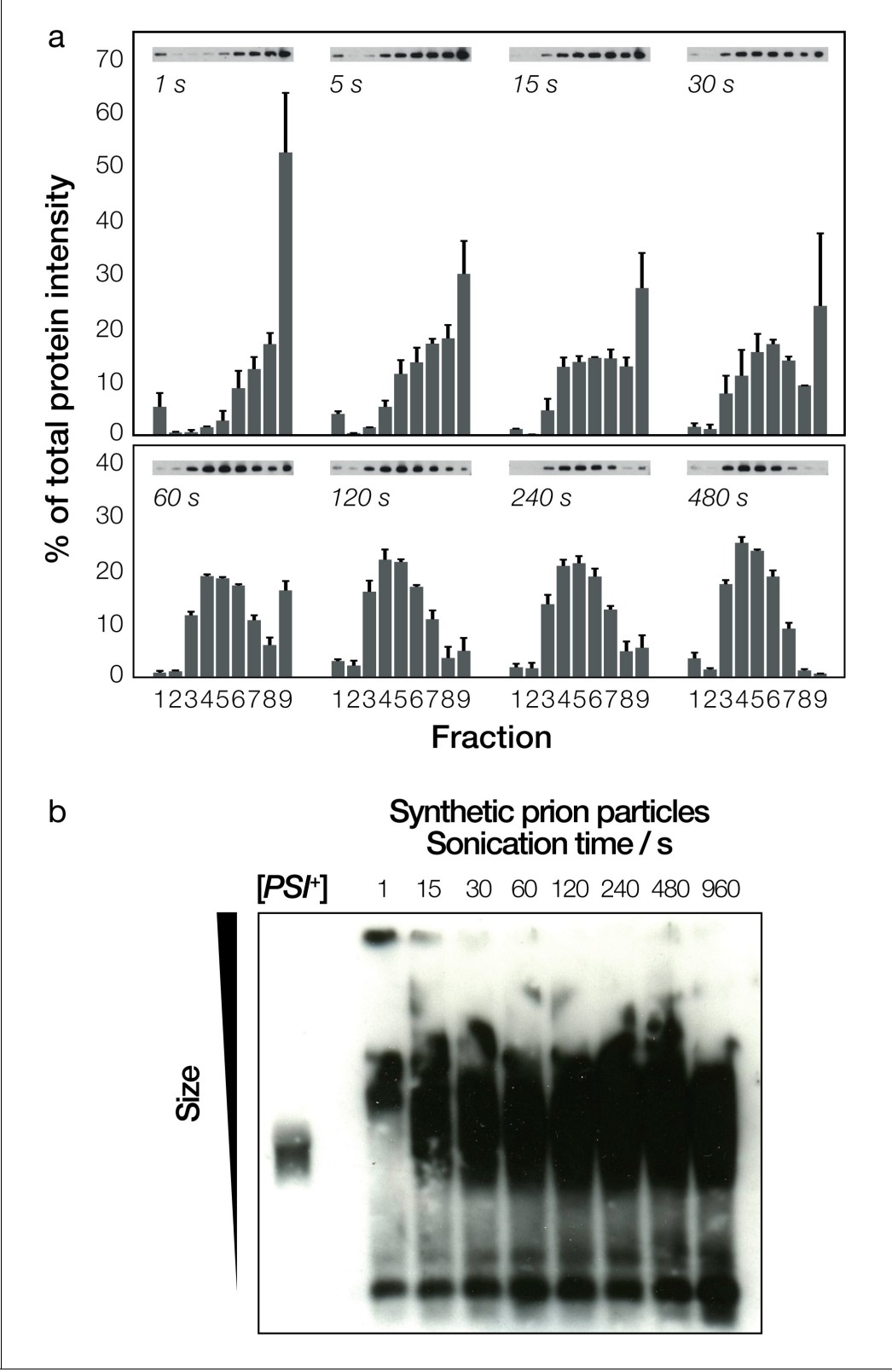

**Figure 3.** Biochemical analysis of fragmented Sup35NM fibrils and their sizes. (a) Native sucrose density gradient analysis of sonicated Sup35NM fibrils. Each group of bars corresponds to one time-point, showing the percentage of total protein present in each of the nine gradient fractions. For each time point, three independent replicates were averaged. Error bars represent standard error of the mean. A representative Western blot for each of the
*Figure 3 continued on next page*

*Figure 3 continued*

time points is also shown. (**b**) SDD-AGE analysis, followed by immunoblotting, of the Sup35NM fibril samples. Samples sonicated for different periods of time are shown in each lane. A cell extract of [*PSI*⁺] cells is shown on the left most lane for comparison.

DOI: https://doi.org/10.7554/eLife.27109.004

The following figure supplement is available for figure 3:

**Figure supplement 1.** Prion transfection efficiency of [*PSI*⁺] cell extract.

DOI: https://doi.org/10.7554/eLife.27109.005

particles when samples were sonicated for longer periods. As seen in *Figure 4a*, the average length of the Sup35NM prion particles reduced from over 210 ± 21 nm to 75 ± 5 nm over the course of 16 min (960 s) of controlled sonication. We also analyzed the height across all particles, which is indicative of the particle widths (*Figure 4c*). This showed that the average particle width remained largely unchanged when sonication time was increased, with the mean height observed at 7.1 ± 0.5 nm. The average dimensions of the fibril particles after extended periods of mechanical perturbation by sonication is also comparable to that expected for protein filaments with tensile strength of 0.2 GPa

**Table 1.** Quantitative analysis of Sup35NM amyloid fibril samples.
Sample and image analysis statistics for characterized Sup35NM amyloid fibril samples.

| Sample | Fragmentation | AFM image analysis | | | | | Prion transfection |
|---|---|---|---|---|---|---|---|
| | Sonication time/s | Number of images | Mean particle length/nm | Number of fibril particles | Mean particle height/nm | Number of pixels | [*PSI*⁺] transfection efficiency / % |
| 1 | 15 | 5 | 189.7 | 1805 | 7.0 | 27965 | 12.3 |
| 2 | 30 | 5 | 121.9 | 6110 | 6.1 | 64381 | 25.0 |
| 3 | 60 | 2 | 112.3 | 5500 | 6.2 | 63114 | 29.4 |
| 4 | 120 | 2 | 111.7 | 5814 | 6.9 | 79594 | 35.0 |
| 5 | 240 | 2 | 88.4 | 7478 | 7.2 | 88175 | 36.7 |
| 6 | 480 | 2 | 74.5 | 8564 | 7.8 | 91816 | 59.5 |
| 7 | 960 | 2 | 77.2 | 8604 | 8.1 | 92713 | 68.9 |
| 8 | 15 | 4 | 231.1 | 1560 | 6.7 | 24809 | 12.6 |
| 9 | 30 | 4 | 141.8 | 2699 | 6.5 | 33303 | 17.3 |
| 10 | 60 | 2 | 184.8 | 2003 | 7.2 | 31931 | 9.2 |
| 11 | 120 | 2 | 125.9 | 4562 | 7.1 | 63937 | 27.7 |
| 12 | 240 | 2 | 84.3 | 4169 | 7.0 | 48486 | 24.7 |
| 13 | 480 | 2 | 68.9 | 5662 | 7.5 | 57393 | 42.0 |
| 14 | 960 | 2 | 70.9 | 5717 | 7.8 | 62894 | 51.2 |
| 15 | 30 | 4 | 142.3 | 675 | 7.1 | 9353 | 6.7 |
| 16 | 60 | 3 | 131.8 | 1937 | 6.8 | 24348 | 13.1 |
| 17 | 120 | 2 | 161.0 | 1909 | 7.1 | 27013 | 23.5 |
| 18 | 240 | 2 | 87.0 | 4269 | 7.0 | 49000 | 21.9 |
| 19 | 480 | 3 | 88.8 | 2938 | 7.0 | 33342 | 21.6 |
| 20 | 960 | 2 | 85.7 | 4687 | 7.4 | 56457 | 35.6 |
| 21 | 30 | 6 | 160.5 | 3306 | 6.9 | 47476 | - |
| 22 | 60 | 4 | 150.0 | 3523 | 7.5 | 47590 | - |
| 23 | 120 | 2 | 184.2 | 2198 | 7.0 | 32565 | - |
| 24 | 240 | 2 | 133.3 | 1803 | 7.5 | 24158 | - |
| 25 | 480 | 2 | 150.6 | 1455 | 7.3 | 21163 | - |
| 26 | 960 | 2 | 64.3 | 8000 | 7.3 | 77914 | - |

DOI: https://doi.org/10.7554/eLife.27109.006

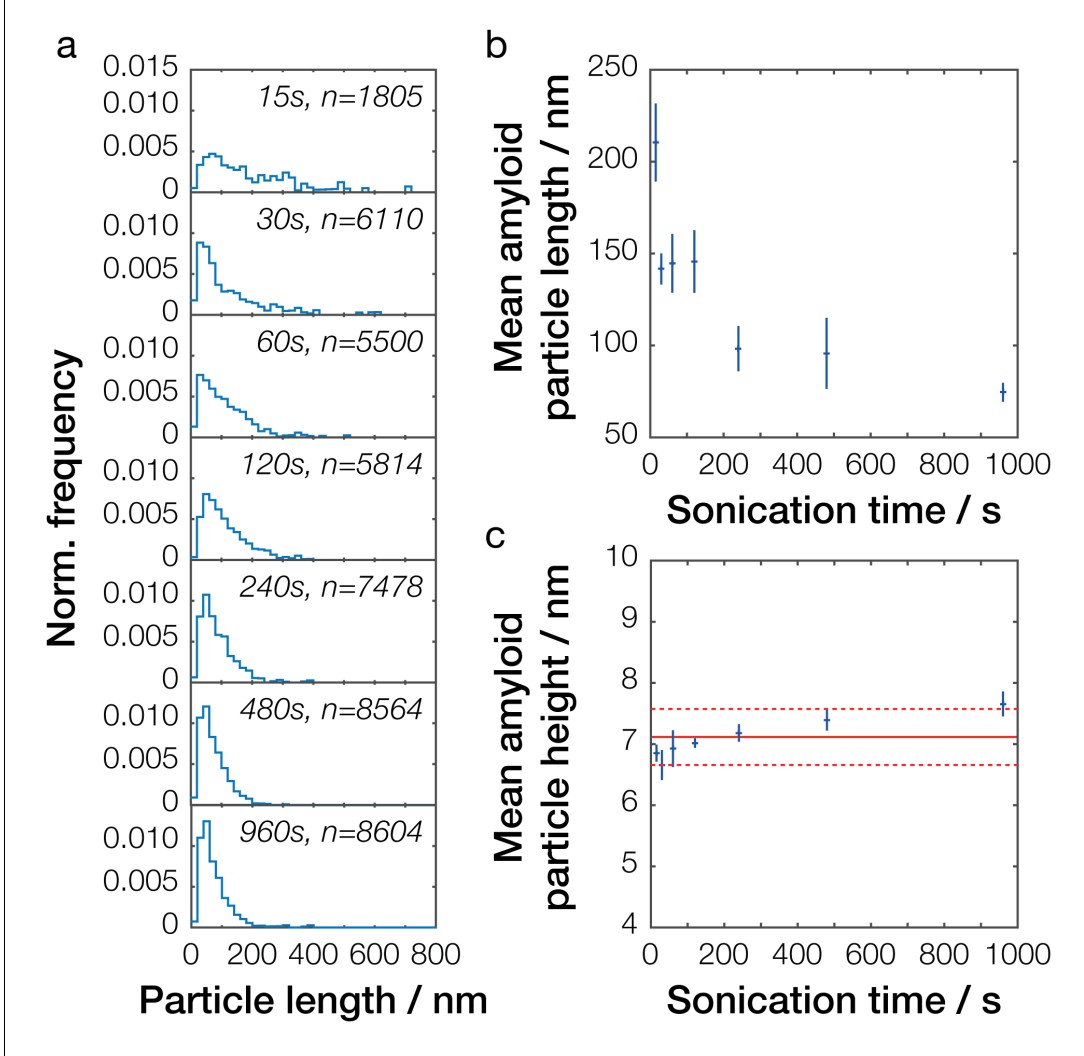

**Figure 4.** Single-particle analysis of Sup35NM fibril length distribution after controlled sonication. (a) Particle size distributions for seven representative Sup35NM samples sonicated for different times. The occurrence of different particle sizes was normalized against the total number of particles traced for each individual sample and plotted against particle length (blue lines). Sonication time and the number of fibrils analyzed for each sample are displayed in each plot. (b) Relationship between mean particle length and sonication time. Each data point represents the mean of all individual samples analyzed for a given time point. Error bars represent the standard error of the mean. (c) Relationship between mean particle height, representing the width of the fibril particles, and sonication time. Each data point represents the mean of all individual samples analyzed for a given time point. Error bars represent the standard error of the mean. The mean height of all values is represented by the solid red line, with its standard error represented by the dotted red lines.

DOI: https://doi.org/10.7554/eLife.27109.007

The following figure supplement is available for figure 4:

**Figure supplement 1.** Particle length distributions for individual Sup35NM samples analyzed by AFM image analysis.

DOI: https://doi.org/10.7554/eLife.27109.008

undergoing sonication-induced scission (*Huang et al., 2009*). These observations are consistent with the fact that the controlled mechanical perturbation resulted in a reduction in particle length, but did not otherwise alter the individual fibril assemblies.

Taken together with the biochemical characterizations of the fibril samples, our results indicate that non-sonicated in vitro generated Sup35NM amyloid particles form a suprastructure consisting of large fibril networks that do not reflect the size and the suprastructure of prion particles present in vivo in [*PSI*⁺] cells. The controlled sonication alters this suprastructure by initially dispersing the fibril network into smaller clustered aggregates, and subsequently produces dispersed fibril particles

with size distributions overlapping with that of particles present in vivo in [*PSI*⁺] cells. Further sonication then proceeds to further reduce the length distribution of the resulting dispersed fibril particles, but the mechanical perturbation employed did not otherwise change the width of these particles.

## Influence of fibril particle concentration and size on prion transfection efficiency

Lastly, we measured the ability of the synthetic fibril samples to induce the [*PSI*⁺] phenotype in vivo in yeast cells. *S. cerevisiae* (74D-694 [*psi*⁻]) cells were transfected by 20 different fibril samples that had their size distributions characterized in detail by AFM image analysis as described above (*Table 1*, *Figure 4—figure supplement 1*). The fibril samples were added to the yeast transfection reaction at the same time they were deposited on mica for the AFM analysis to eliminate the impact of sample-to-sample particle size variations (as shown in *Figure 4*) on [*PSI*⁺] transfection efficiency determinations. *Figure 5a and b* show the relationship between the average particle lengths of the samples and their efficiencies in inducing the [*PSI*⁺] phenotype. Under the conditions employed, fibril samples with average particle length ranging from 231 nm to 69 nm resulted in [*PSI*⁺] transfection efficiencies ranging between 7% and 69% (*Table 1*).

Knowing the length distribution of the fibril samples allows the calculation of their fibril particle concentrations (*Xue and Radford, 2013*). Comparing [*PSI*⁺] transfection efficiencies with their corresponding fibril particle concentrations (*Figure 5b*) revealed a striking correlation, indicating that the ability of Sup35NM fibril particles to induce the [*PSI*⁺] prion phenotype depends on the number of particles present. Interestingly however, as seen in *Figure 5b*, this correlation deviated from the expected linear behavior where 0% [*PSI*⁺] prion phenotype would result from a particle concentration of 0 M (the x-intercept of the dashed line is 12.2 nM in *Figure 5b* inset), suggesting that fibril particles are not equally capable at inducing the [*PSI*⁺] prion phenotype. This is consistent with the fact that fibril samples that were not sonicated (*Figure 2b*, *upper left image*) cannot induce the [*PSI*⁺] prion phenotype, since the particle concentration for the non-sonicated sample still must be greater than 0 M. Therefore, our analysis suggests that longer particles are less able to induce the [*PSI*⁺] prion phenotype compared to identical numbers of their shorter counterparts.

There are two possibilities that can explain the observed higher infective potential of short particles compared with the same number of their longer counterparts; (a) the particles of different lengths are not equal in recruiting soluble intracellular Sup35 and seeding the conversion and growth of new amyloid, or (b) the particles are not equally capable of crossing the cell membrane and access the cellular environment to elicit the [*PSI*⁺] phenotype. To test whether the observed differences were due to a poor ability of larger particles to seed amyloid growth, we performed seeded amyloid growth assays monitored by Thioflavin T (*Figure 5—figure supplement 2*) where 10 uM fresh Sup35NM monomers were seeded by 1% of the fibril samples sonicated to different extents. The seeding potential of the fibril particles was determined by measuring the initial slope of the seeded growth curves and the resulting slope value compared with each sample's known particle concentration. All samples analyzed were capable of seeding the polymerisation of monomeric Sup35NM, albeit to different extents, with those sonicated for longer periods doing so with greater efficiency (*Figure 5—figure supplement 2a*) as expected for amyloid growth by monomer addition at fibril ends. As seen in *Figure 5—figure supplement 2b* (blue data points), and in contrast with the [*PSI*⁺] transfection efficiencies, the seeding ability of the different fibril samples depended solely on the number of fibril particles, with the initial slope of seeded growth curves showing a linear correlation with particle concentration that was not significantly different to the expected trend where 0 M particles should result in 0 hr$^{-1}$ initial slope. To further confirm this result, we made a prediction that seeding with the same particle concentrations using samples of different length distributions should result in the same seeding efficiency. Here, the average length of particles in sample sonicated for 60 s is 145 nm, about twice as long as average length of particles in sample sonicated for 960 s (average length of 75 nm). Therefore, the particle concentration of the sample sonicated for 60 s is roughly half that of the sample sonicated for 960 s. Consequently, an assembly reaction seeded by 2% of the sample sonicated for 60 s is predicted to be as efficient as the reaction seeded by 1% of sample sonicated for 960 s. We carried out experiments to test this prediction by seeding new reactions with 1% as well as 2% of an independent fibril sample sonicated for 60 s (*Figure 5—figure supplement 2b*, yellow data points). As seen in *Figure 5—figure supplement 2b*, a reaction seeded by 2% of the independent sample sonicated for 60 s (upper right yellow cross) reproduced

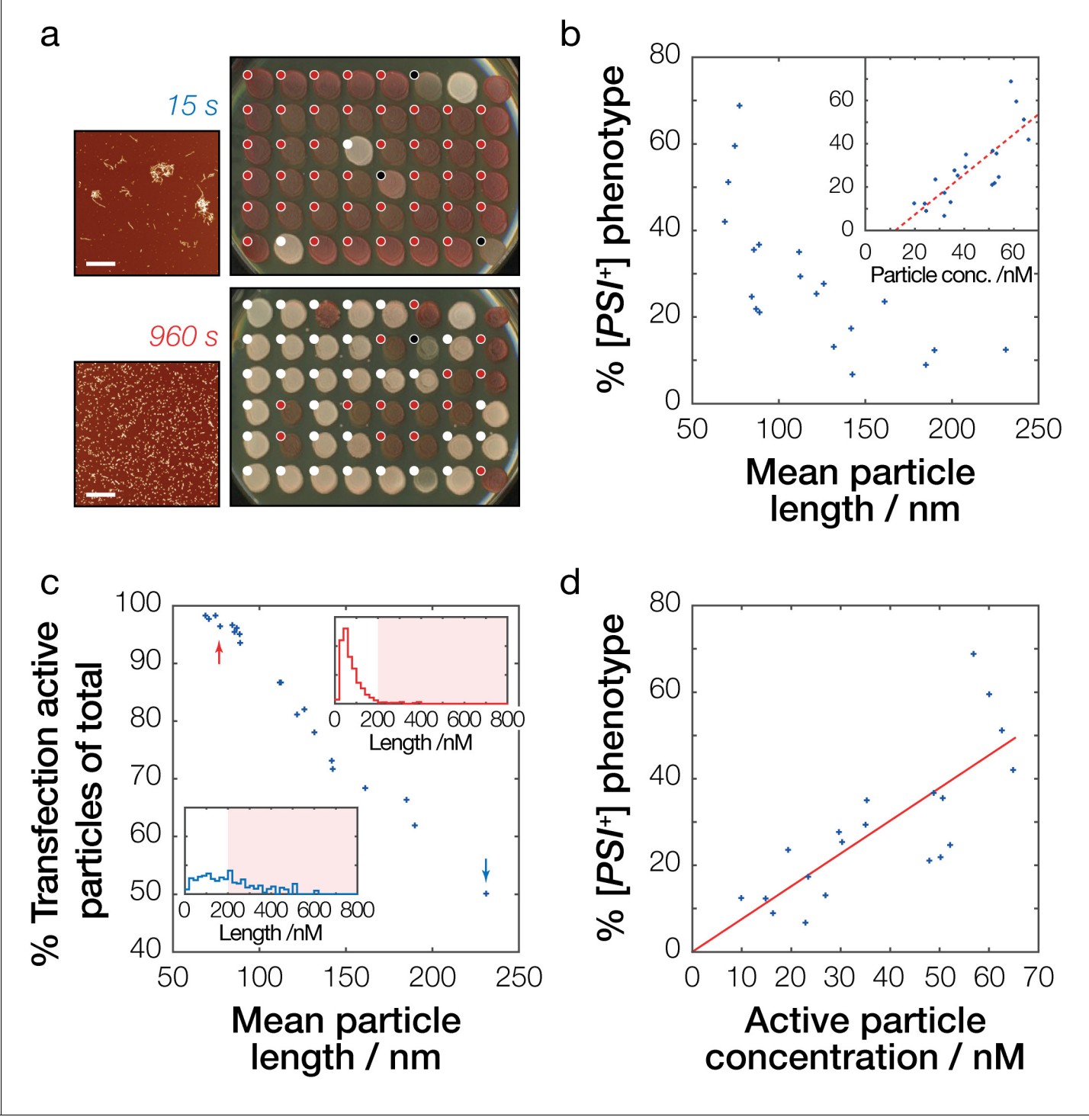

**Figure 5.** Prion infective potential depends on particle size, concentration and activity. (a) Analysis of prion transfection efficiency. AFM images of representative Sup35NM fibrils samples sonicated for 15 s (upper) and 960 s (lower) are shown together with plates containing yeast colonies grown from protoplasts transfected with the Sup35NM fibrils samples, indicating the sample's ability to infect yeast cells and to induce [*PSI*⁺] phenotype in vivo. Scale bars indicate the length of 1 μm. Individual yeast colonies were scored as [*PSI*⁺] (white dots) or [*psi*⁻] (red dots) based on their colour in ¼ YEPD and curability in ¼ YEPD supplemented with 3 mM GdnHCl (*Figure 5—figure supplement 1*). Colonies that showed poor growth or unrecognisable colour differentiation were omitted (black dots). On each plate, control [*PSI*⁺] (white) and [psi⁻] (red) colonies are present at the upper right corners for comparison. See *Figure 5—figure supplement 1* for the full data set. (b) Dependency of prion transfection efficiency on particle size. Mean fibril length for each of the 20 samples analysed is plotted against the percentage of *S. cerevisiae* [*PSI*⁺] colonies obtained after prion

*Figure 5 continued on next page*

*Figure 5 continued*

transfection. Inset show dependency of prion transfection efficiency on particle concentration calculated from the length distribution of the samples. The dashed red line denotes best fit linear model with the slope of $9.3 \cdot 10^8$ M$^{-1}$ and the intercept of $-11.3\%$ with 95% confidence interval for the intercept between $-26.0\%$ to $-0.7\%$, demonstrating that this linear model is not the correct model to describe the transfection efficiency as function of particle length or particle concentration. (c) Transfection activity of the Sup35NM fibrils samples estimated as the percentage of fibril particles less than 200 nm long show against the average particle length of each fibril sample. Insets show representative particle length distributions of the fibril samples shown in (a) sonicated for 15 s (blue distribution/arrow) and 960 s (red distribution/arrow), with shaded areas denoting particles larger than 200 nm long. (d) Dependency of prion infective potential on active particle concentration consisting of fibril particles less than 200 nm long. The red line denotes best-fit linear model with the slope of $7.6 \cdot 10^8$ M$^{-1}$.

DOI: https://doi.org/10.7554/eLife.27109.009

The following figure supplements are available for figure 5:

**Figure supplement 1.** Prion transfection efficiency of Sup35NM amyloid fibrils samples.

DOI: https://doi.org/10.7554/eLife.27109.010

**Figure supplement 2.** Seeding efficiency of Sup35NM amyloid fibrils samples.

DOI: https://doi.org/10.7554/eLife.27109.011

**Figure supplement 3.** Testing model predictions on the prion transfection efficiency of Sup35NM amyloid fibrils samples of different length but the same active concentration.

DOI: https://doi.org/10.7554/eLife.27109.012

the same efficiency as a reaction seeded with 1% sample sonicated for 960 s (upper right blue cross) that had a comparable particle concentration. These results rule out that the particles are sufficiently unequal in seeding the conversion and growth of new amyloid, and therefore suggest that particles are not equally capable of crossing the cell membrane to access the intracellular environment and elicit the [*PSI*$^+$] phenotype.

Next, we investigated how particle size might modulate the relationship between particle concentration and [*PSI*$^+$] transfection efficiencies. Were transfection efficiency dependent solely on particle concentration, it would be expected for a transfection efficiency of 0% to occur at 0 M particle concentration and increase linearly from that point. This was not the case for our data (dashed line in *Figure 5b*). Hence, we propose the introduction of a transfection activity coefficient, $\gamma_{transf}$, that is capable of representing the fibril particles' infective potential. We then define an active particle concentration $c_{p,transf}(l)$ depending on the particle length $l$ so that:

$$c_{p,transf}(l) = \gamma_{transf}(l) \cdot c_p(l) \tag{1}$$

where $c_p$ is the particle concentration and $l$ is particle length. We then assume the simplest possible model where there is a particle size 'cut-off' $l^*$, and particles longer than this cut off will not be able to transfect yeast cells and induce the [*PSI*$^+$] prion phenotype (i.e. $\gamma_{transf}$ for an individual particle is 0 when its length is longer than $l^*$ and one if its length is shorter or equal than $l^*$). This can be written as the following relationships:

$$\gamma_{transf}(l, l^*) = \begin{cases} 1, & l \leq l^* \\ 0, & l > l^* \end{cases} \tag{2}$$

The total transfection active particle concentration $c_{p,transf}$ is then the sum of all active particles:

$$c_{p,transf} = \sum_l c_{p,transf}(l) = \sum_l \gamma_{transf}(l, l^*) \cdot c_p(l) \tag{3}$$

To establish the particle size 'cut-off' $l^*$ that is most consistent with our data, we systematically tested possible $l^*$ values, and found that when $l^*$ is 200 nm (*Figure 5c*) then the calculated activity of the fibril samples in terms of their active particle concentration satisfies the criteria that it correlates with the transfection efficiency with the anticipated transfection efficiency of 0% occurring at 0 M particle concentration (*Figure 5d*). To test the predictive abilities of this model, we next calculated the average active particle concentration of the whole sample sonicated for 15 s and 960 s, respectively. For the sample sonicated for 15 s, the particle concentration was estimated to be 22 nM based on their average length of 210 nm, and the average transfection activity coefficient of this sample was 0.55 (*Figure 5c*). According to our model with $l^* = 200$ nm, this gives for transfection an active particle concentration of 12.1 nM. For the sample sonicated for 960 s, the particle

concentration was estimated to be 61 nM from average length of 75 nm, and the average transfection activity coefficient of this sample was 0.98 (*Figure 5c*), giving an active particle concentration of 59.8 nM, roughly five times higher than the sample sonicated for 15 s. Consequently, our model predicts that a five times dilution of a sample sonicated for 960 s will result in the same transfection efficiency as a sample sonicated for 15 s. On the other hand, if our size cut-off model is incorrect and all effect on transfection efficiency is based purely on the total particle concentration then we expect five times dilution of a sample sonicated for 960 s will give roughly half the transfection efficiency compared to that of a sample sonicated for 15 s. We tested this prediction experimentally by measuring the ability of independent samples sonicated for 15 s and 960 s to induce the [*PSI*⁺] phenotype (*Figure 5—figure supplement 3*). This experiment confirmed that the transfection efficiencies of the samples sonicated for 15 s is indistinguishable compared to five times diluted samples sonicated for 960 s, which was predicted to have equal active concentration of prion particles. Thus, these results are consistent with the predictions of our size cut-off model that particles longer than 200 nm in length are likely to be incapable of entering the cells and induce the [*PSI*⁺] prion phenotype. In reality, the efficiency with which fibril particles enter the cells and propagates the prion phenotype may be a continuous and non-linear function of their dimensions. However, our simple model, with a size cut-off estimate that particles of 200 nm or less in length are capable of entering the cells and conferring the [*PSI*⁺] prion phenotype, is fully consistent with the data. Assuming the fibril volume is estimated by cylinders of 7.1 nm diameter based on the AFM height data (*Figure 4c*) and a fibril density comparable to that of folded proteins at 1.4 g/cm³ (*Fischer et al., 2004*), then the molecular weight of a 200 nm particle is approximately 7 MDa. The estimated particle length cutoff is also corroborated by the AFM images of the sample series, showing fibril clusters persist in the same size range as our cutoff estimate or larger (e.g. arrows in 15, 30 and 60 s images in *Figure 2b*). In summary, these results demonstrate that the infective potential of the Sup35NM prion samples depends on effective particle concentrations that only take into consideration a subpopulation consisting of prion particles of optimal dimensions for transfection.

## Discussion

Infectivity and cell-to-cell propagation are two of the main criteria that set prions apart from other amyloid aggregates (*Tuite and Cox, 2003*). Amyloid fibril fragmentation is a crucial process that potentiates propagation by increasing the number of transmissible, seeding competent particles and as we demonstrate here, also by producing particles that are of an 'ideal size' for transmission. A growing number of disease-associated amyloid forming proteins appear to possess prion-like properties in that these amyloid particles can be transmitted to nearby cells to effectively propagate the amyloid-associated phenotype to previously healthy cells (*Aguzzi and Lakkaraju, 2016*). These findings blur the line between transmissible and non-transmissible amyloid, suggesting that the infectious potential of amyloid is a complex biological property that is better described by a sliding scale rather than the binary prion/non-prion view.

In the case of the prion-like amyloid proteins such as Aβ (*Nussbaum et al., 2012*), α-synuclein (*Masuda-Suzukake et al., 2013*), tau (*Sanders et al., 2014*), and huntingtin (*Pearce et al., 2015*), transmissibility has been linked to a number of active protein import and export mechanisms (*Aguzzi and Lakkaraju, 2016*). However, the possibility that cell-to-cell transmission occurs by diffusion across the cellular membrane cannot be discounted. Studies of the mammalian prion protein PrP have shown that small prion particles consisting of 14–28 PrP monomers are more infections than their larger counterparts (*Silveira et al., 2005*), indicating that particle size plays an important role in mammalian prion infectivity. Furthermore, it has also been shown that exogenous, recombinant Sup35NM amyloid can be used to infect and confer a prion phenotype to mammalian N2a cells expressing a soluble, cytosolic form of Sup35NM (*Krammer et al., 2009*). Taken together, these data indicate that transmissibility might be a general property of all amyloid aggregates, which will invariably occur given the right physical properties and conditions. This makes it crucial that we fully understand how the mesoscopic and suprastructural properties of amyloid particles influences their transmissibility as well as identifying how passive and/or active protein transport mechanisms could contribute to this phenomenon.

Once formed in a cell, the continued and efficient propagation of a given yeast prion occurs as cells divide, fuse during mating (*Tuite and Cox, 2003*) or give rise to the products of meiosis

(sporulation), and is greatly facilitated by the cytoplasmic location of the transmissible forms of the prion. This propagation is thought to occur passively by cytoplasmic transfer, as no active mechanisms for transmission of prion particles have yet been identified (*Byrne et al., 2009*) although the possibility that extracellular vesicles may facilitate the vertical and horizontal transmission of yeast prions has been raised (*Kabani and Melki, 2015*). Infection with amyloid particles can also be achieved experimentally by transfecting yeast protoplasts that are largely devoid of the normally protective and robust cell wall (*King and Diaz-Avalos, 2004*; *Tanaka et al., 2004*).

In this study, we have taken advantage of the fact that we could quantitatively determine the lengths of single prion particles using AFM image analysis and calculate particle concentrations. By then coupling this with yeast transfection we have been able to determine how these properties affected the efficiency with which they crossed the yeast cell membrane into the cytoplasm and induce the [*PSI*⁺] prion phenotype in vivo. Use of this now well-established yeast prion infection model has allowed us to systematically investigate how length distribution and particle concentration affect the activity of amyloid particles to cross cellular membranes and infect yeast cells. The key conclusion that has emerged from our analysis is that infectivity is only proportional to particle concentration when the particles are of favorable size and free from forming aggregate suprastructures (*Figure 6*).

Using a simple model to estimate the infectious activity of Sup35NM prion samples based on their length distribution, we show that the particle concentration versus transfection activity correlation only applies when taking into account an active particle concentration based on prion particles up to a certain length. This has led us to estimate the size cut-off for infectivity of Sup35NM particles at approximately 200 nm. Above 200 nm, the individual Sup35NM prion particles are likely to interact and form higher-order aggregate suprastructures (*Figure 2b*, 0 s, and arrows in 15–60 s), which bear similarity to what has been described as flocculation or gelation (*Buell et al., 2014*). The fibrils in the aggregated suprastructures are relatively inert compared with their shorter, less aggregated counterparts, and they are non-infectious. Below this size threshold, infectivity is largely dependent on particle concentration. Thus, the infectious potential of prion samples, in terms of average transfection activity per particle, is likely a non-linear function (e.g. *Figure 5c* for Sup35NM particles analyzed here) that depends on the size distribution of the particles (*Figure 6*). The activity of the particles as function of their size will subsequently depend on particle-particle interactions coupled with the mesoscopic properties of the aggregates, and may be further dependent on the particles' interactions with other cellar components inside cells, on their interactions with membranes and other surfaces, and on their diffusion properties. In terms of surface interactions, the same active surface that promotes secondary nucleation may also give rise to particle-particle interactions or interactions with membranes and other surfaces (*Buell et al., 2014*), thus modulating the suprastructure and infective potential of the particles. In the case of diffusion, the effect of particle size on translational and rotational diffusion coefficients of rod-like particles are proportional to 1/length and 1/length$^3$, respectively (*Ortega et al., 2003*). Therefore, the length dependence of diffusion is considerably larger for small particles roughly below ~50 nm in length compared with their longer counterparts (*Ortega et al., 2003*), suggesting that diffusion may play additional role for the increased activity of small mobile particles if they are much smaller than ~50 nm. In any case, resolving the activity function (*Equation 2*) for other prions and prion-like amyloid systems and understanding the molecular origins of the constitutive components of this function in order to understand the size–suprastructure–activity relationships of the amyloid and prion particles will undoubtedly reveal why some amyloid aggregates are inert while others are cytotoxic and/or infectious.

Successful prion infection depends not only on a particle's ability to cross the cellular membrane but also on its interactions with intracellular factors such as chaperones, co-chaperones and the proteostasis machinery in general. These downstream interactions are eventually translated into successful propagation of prions in yeast and in pathogenic prion systems. While our data suggest that the size threshold and clustering and fibril network formation result in a reduced ability of particles to cross the cell membrane, further direct observations of prion particles entering the cells and interacting with cellular machineries inside the cell volume will shed light on the role of intracellular processes such as chaperone interactions and cellular sequestration or degradation, which may also influence the effect of particle shape, size and suprastructure on prion propagation. Much like the data we here show for prion infectivity, intracellular prion propagation in yeast has previously also

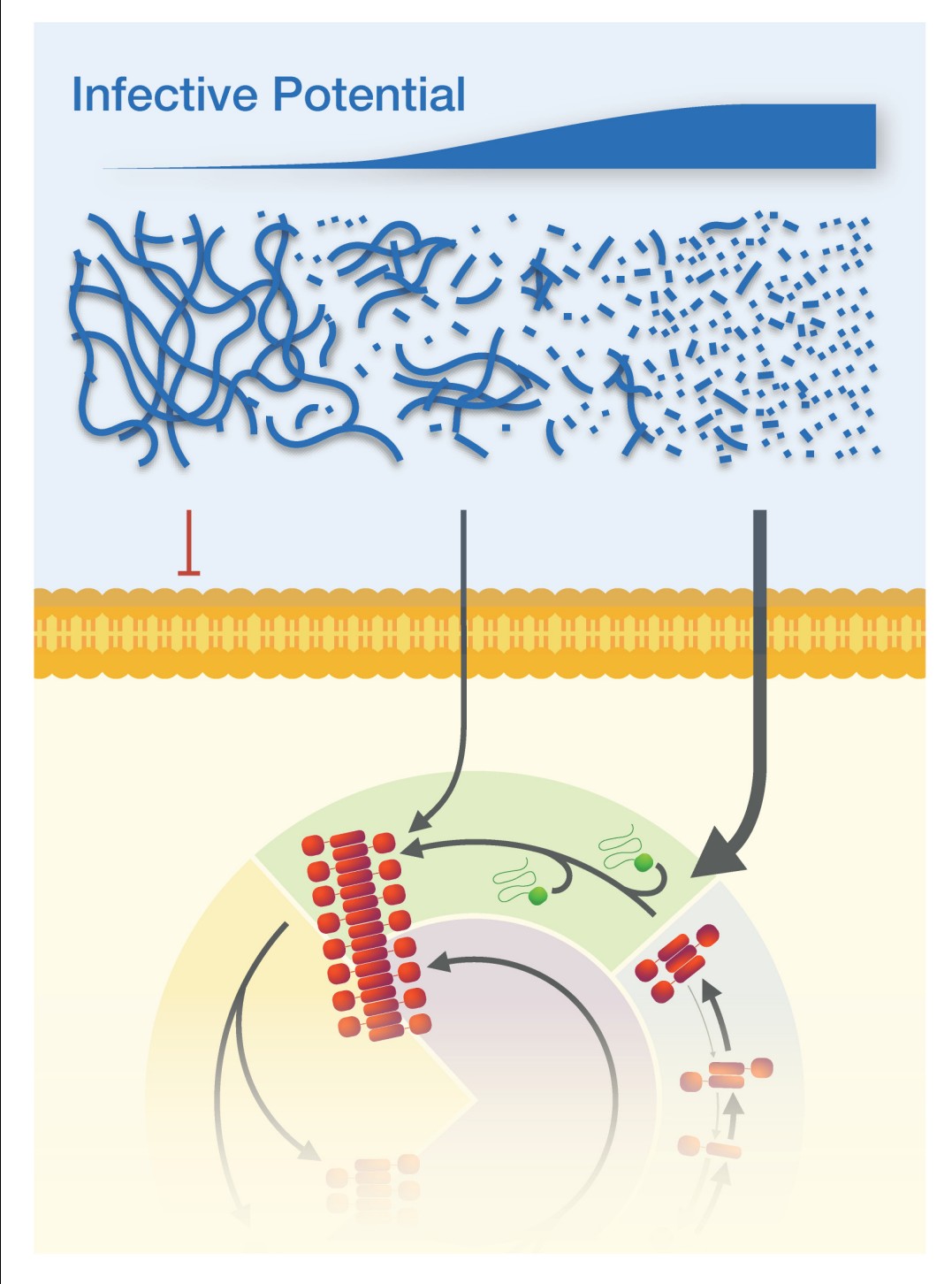

**Figure 6.** The physical dimensions of prion particles modulate their suprastructure and their infective potential. The length of individual Sup35NM amyloid fibrils modulate whether they are active prion particles that can infect cells and induce the [*PSI*⁺] prion phenotype or form inactive suprastructures in the form of fibril clusters and networks. Thus, the infective potential of prion samples may be represented in terms of an average per particle activity (top sliding bar). Here, the active Sup35NM yeast prion population is represented by an active population of particles less than ~200 nm long, which are able to cross the membranes of yeast protoplasts and enter the prion lifecycle in the cells. Particle sizes larger than 200 nm are non-infectious, most likely because they cannot cross the cell membrane, but this may also be due to the possibilities that they cannot interact with or are actively sequestered away from the propagation machinery due to their aggregate size.

*Figure 6 continued on next page*

*Figure 6 continued*

DOI: https://doi.org/10.7554/eLife.27109.013

been shown to be a 'size-based' process (*Derdowski et al., 2010*), which further emphasizes existence of physical size barriers in amyloid transmission.

In conclusion, our findings show that the physical dimensions are an important parameter to take into account when assessing the infective potential of amyloid particles. Since large amyloid fibril networks are likely to be relatively inert suprastructures compared with small diffusible fibril particles, calculating size thresholds for effective transmission activity should be of particular importance when examining the infectious potential of prion-like amyloid structures as well as large structures such as amyloid plaques, which are thought to be largely inert (*Tipping et al., 2015*). This leads us to suggest that strategies to promote pre-formed amyloid assembly into large, non-transmissible amyloid structures might be able to control their prion-like transmission, and subsequently delay amyloid disease progression. Further analysis of the size and infective potential of the large amyloid suprastructures as well as smaller amyloid polymers or particles, which might arise through mechanical or enzymatic action (*Powers et al., 2012*), will also be crucial in deciphering their fundamental biological and cellular activities. A molecular understanding of the amyloid fibrils' stability towards fragmentation, as well as how fibril fragmentation is catalyzed in the cell by chaperones such as Hsp104, Hsp70 and Hsp40, will lead us to a better understanding, and suggest diagnostics and therapeutics to combat amyloid transmission, amyloid disease progression and prion infectivity.

# Materials and methods

## Protein expression and purification

The region of the *SUP35* gene encoding the NM region of the yeast Sup35 protein (residues 1–253) was amplified from plasmid pUKC1620 by PCR and cloned into pET15b as a *Bam*HI-*Nde*I fragment. This generated an N-terminal His$_6$-tag fusion protein. The resulting plasmid (pET15b-His$_6$-NM) was then transformed into the *E. coli* strain BL21 DE3 (*F– ompT gal dcm lon hsdSB(rB- mB-) λ(DE3 [lacI lacUV5-T7 gene 1 ind1 sam7 nin5]*). For protein expression, BL21 DE3 was grown overnight in 50 ml LB supplemented with 0.1 mg/ml ampicillin and then transferred to 1 L cultures of the same medium. On reaching an OD$_{600}$ of approximately 0.5, expression was induced with 1 mM IPTG for 4 hr. Cells were harvested at 6000 rpm and the cell pellets washed once in buffer A1 (20 mM Tris-HCl pH 8.0, 1 M NaCl, 20 mM Imidazole). Cells were pelleted and kept at −80°C for later use. For the affinity purification step, buffer A2 (20 mM Tris-HCl pH 8.0, 1 M NaCl, 20 mM Imidazole, 6 M GdnHCl) was added to the frozen cell pellets at a 5:1 (v/v) ratio, followed by sonication at an amplitude of 22 microns until the cell pellet was completely disrupted. This solution was then subject to centrifugation at 13000 rpm for 30 min and the resulting supernatant collected. 1 ml of Chelating Sepharose Fast Flow (GE Healthcare) was added to a small plastic column and prepared for affinity purification by sequential washing with one column volume (CV) of water, 0.2 M NiCl$_2$, buffer A1 and buffer A2. The equilibrated resin was then resuspended in buffer A2 and added to the previously collected supernatant. This mixture was then incubated for 1 hr at room temperature with agitation to improve protein binding to the affinity resin. Centrifugation at 5000 rpm was subsequently used to collect the resin, which was then washed in 5 ml buffer A2 and resuspended in buffer A2 and transferred back to the column. After one wash with 1 CV buffer A2, elution was achieved by addition of 4 ml buffer A3 (20 mM Tris-HCl pH 8.0, 1 M NaCl, 0.5 M Imidazole, 6 M GdnHCl). The resulting eluate was immediately used in size-exclusion purification, which was run using a HiLoad 16/600 Superdex 200 pg (GE Healthcare) column in an ÄKTA Prime Plus chromatography system (GE Healthcare). The eluate was injected into the size-exclusion column previously equilibrated with 1 CV water followed by 1 CV buffer S1 (20 mM Tris-HCl pH 8.0, 0.5 M NaCl) and 1 CV buffer S2 (20 mM Tris-HCl pH 8.0, 0.5 M NaCl, 6 M GdnHCl). The relevant Sup35NM protein fractions were collected according to the A$_{280}$ displayed throughout the run, diluted to 20 µM in buffer S2 and immediately used in fibril-forming reactions.

## Sup35NM fibril formation

For fibril formation, 2.5 ml of 20 µM purified Sup35NM were buffer exchanged into Fibril Forming Buffer (FFB - 20 mM $Na_2PO_4$ pH 7.4, 50 mM NaCl) using a PD-10 column (GE Healthcare) as per manufacturer's instructions. Protein concentration was measured using $A_{280}$ and then adjusted to 10 µM using FFB. Protein was aliquoted into Protein LoBind tubes (Eppendorf) and polymerized at 30°C (quiescent) for at least 48 hr. For monitoring polymerisation, 100 µl samples of protein were aliquoted into black puregrade 96-well plates (BRAND) and Thioflavin T was added to a final concentration of 10 µM. The plate was sealed with Starseal Advanced Polyolefin Film (Starlab) and kinetics were monitored in a FLUOstar OMEGA plate reader (BMG Labtech) at 30°C.

## Sup35NM fibril fragmentation

Sup35 fibril samples were concentrated by centrifugation at room temperature (13000 rpm, 40 min) and resuspended in 1:10th of the volume of FFB, unless stated otherwise. Fibril fragmentation was achieved by sonication over different periods using a probe sonicator (Qsonica Q125) at 20% amplitude in consecutive 5 s on/off cycles on ice cooled water-bath.

## Sucrose density gradient analysis

15% to 60% sucrose gradients were prepared in FFB. Fibril samples used in sucrose gradients were concentrated by centrifugation at room temperature (13000 rpm, 40 min) and then resuspended in half the volume of FFB before sonication, which was performed as described above. 100 µl of samples sonicated for different amounts of time were pipetted onto the top of a sucrose gradient and centrifuged at 31,000 rpm, 4°C, for 3 hr. Gradients were sampled while on ice by pipetting 221 µl fractions from the top of the gradient into pre-chilled tubes which were snap-frozen in liquid nitrogen and stored at −80°C for subsequent western blot analysis. Each gradient produced nine fractions. 10 µl of each fraction was analysed on a 4–20% polyacrylamide Tris-Glycine gradient gel (Invitrogen, Waltham, MA) run at 125 V. Proteins were transferred onto a PVDF membrane by semi-dry blotting (10 V, 45 min) and membranes were probed with anti-Sup35 (MT50) polyclonal antibody. Anti-rabbit HRP-conjugated antibody was used as a secondary antibody in standard ECL analysis. For densitometry, the image analysis software ImageJ (version 1.42, http://rsbweb.nih.gov/ij/, RRID:SCR_003070) was used. Relative intensity of each band was calculated by dividing the given intensity value for each band by the total intensity value for all bands in that sample.

## Semi-denaturing agarose gel electrophoresis (SDD-AGE)

SDD-AGE analysis was performed as previously described (Kryndushkin et al., 2003) with the following modifications. Sonicated fibrils were obtained as described above and loaded in a 1.5% agarose gel prepared in buffer G (20 mM Tris, 200 mM glycine) and ran on Laemmli buffer (20 mM Tris, 200 mM glycine, 0.1% SDS). Proteins were transferred using semi-dry blotting and transfer buffer T (20 mM Tris, 200 mM Glycine, 0.1% SDS, 15% (v/v) methanol) onto a PVDF membrane for 90 min at 10 V. MT50 anti-Sup35 primary antibody was used in western blot analysis as described above. Cell extracts used in SDD-AGE were prepared by first harvesting yeast cells ($\approx 2 \cdot 10^7$ cells) and resuspending the pellet in 100 µl PEB buffer (25 mM Tris-HCl pH 7.5, 50 mM KCl, 10 mM MgCl2, 1 mM EDTA and EDTA-free Protease Inhibitor Cocktail [Roche]). Approximately one pellet volume of small glass beads was added to the resuspended cells and lysis performed by vortexing at 4°C. Lysate was then cleared by centrifugation (8000 rpm, 10 min, 4°C) and total protein concentration in the collected clear lysate was measured by A280. Approximately 100 µg total protein were loaded per lane.

## Prion transfection

For prion transfection with synthetic Sup35NM amyloid fibrils, a [*psĭ*] derivative of the yeast strain 74D-694 (*MATα ade1-14 trp1-289 his3Δ−200 ura3-52 leu2-3,112*) was used with an adaptation of a previously published amyloid fibril transformation protocol (Tanaka, 2010). Cells were inoculated in 5 ml YEPD and grown overnight at 30°C with agitation. They were then diluted into fresh YEPD and allowed to grow to an $OD_{600}$ of 0.5. Cells were than washed and resuspended in 12 ml ST buffer (1 M sorbitol, 10 mM Tris-HCl pH 7.5). Spheroplasts were prepared by addition of 600 U of lyticase (Sigma L4025) and 10 mM DTT and incubated at 30°C with agitation for 45 min. Spheroplasts were then harvested by centrifugation (400 x*g*, 5 min), successively washed with 1.2M sorbitol and STC

buffer (1.2 M Sorbitol, 10 mM Tris-HCl pH 7.5, 10 mM CaCl$_2$) and then resuspended in 1 ml STC buffer. Each transformation reaction consisted of 100 µl spheroplast suspension, 2 µl (approximately 1 µg) of plasmid DNA (pRS416), 10 µl single-stranded DNA (10 mg/ml) and 10 µl of freshly sonicated Sup35NM amyloid fibrils (as described above) or 1 µl of cell extract. This transformation mix was incubated for 10 min at room temperature and then 0.9 ml of PEG buffer (40% PEG 4000, 10 mM Tris-HCl pH7.5, 10 mM CaCl$_2$) was added to each transformation. After 30 min at room temperature, the spheroplasts were collected by centrifugation (400 x$g$, 5 min), resuspended in 200 µl SOS media and 20 µl were added to sterile Top agar (-URA synthetic complete media with 2% agar and 1.2 M Sorbitol) being kept at 48°C, gently mixed and then poured into agar plates previously prepared using the same medium. Cells were allowed to grow for 3–4 days and then colonies were individually picked into 96 well plates containing YEPD. These were grown overnight at 30°C with agitation and then replica plated onto ¼ YEPD to check for the [$PSI^+$] prion phenotype and ¼ YEPD supplemented with 3 mM GdnHCl to eliminate any false positives; 3 mM GdnHCl eliminates the [$PSI^+$] prion (*Tuite et al., 1981*). Fragmented amyloid fibrils used in transformation experiments were simultaneously prepared for particle size distribution analysis using AFM as described below.

### Atomic Force Microscopy

20 µl of the fibril samples were diluted 1:300 and deposited on a freshly cleaved mica disc (Agar Scientific F7013). After 10 min incubation at room temperature, excess sample was removed by washing with 1 ml of 0.2 µm syringe filtered mQH$_2$0 and then dried under a gentle stream of N$_2$. Samples were imaged using a Bruker Multimode AFM with a Nanoscope V controller and a ScanAsyst probe (Silicone nitride tip with tip height = 2.5–8 µm, nominal tip radius = 2 nm, nominal spring constant 0.4 N/m and nominal resonant frequency 70 kHz). Images were captured at a resolution of 4.88 nm per pixel scanned. All images were processed using the Nanoscope analysis software (version 1.5, Bruker). The image baseline was flattened using third order baseline correction to remove tilt and bow, and the data was saved as processed image files and raw data files, for recognition by the fibril tracing software. A larger number of images were collected for low sonication time point samples, as long fibrils are harder to measure due to a tendency to tangle and associate in larger structures (see *Figure 2b*) that dissociate over extended periods of sonication. Processed image files were opened and analyzed using automated fibril tracing scripts written in Matlab (*Xue, 2013*).

## Acknowledgements

We thank the members of the Xue group, and the Kent Fungal Group for helpful comments and discussions, and Ian Brown for technical support. This work was supported by funding from the Biotechnology and Biological Sciences Research Council (BBSRC), UK grants BB/J008001/1 and BB/M02427X/1, as well as BB/F016719/1 (DMB).

## Additional information

### Funding

| Funder | Grant reference number | Author |
|---|---|---|
| Biotechnology and Biological Sciences Research Council | BB/J008001/1 | Ricardo Marchante<br>Tracey J Purton<br>Wei-Feng Xue |
| Biotechnology and Biological Sciences Research Council | BB/F016719/1 | David M Beal |
| Biotechnology and Biological Sciences Research Council | BB/M02427X/1 | Nadejda Koloteva-Levine<br>Tracey J Purton<br>Mick F Tuite<br>Wei-Feng Xue |

The funders had no role in study design, data collection and interpretation, or the decision to submit the work for publication.

## Author contributions

Ricardo Marchante, Conceptualization, Data curation, Formal analysis, Investigation, Methodology, Designed the research, Conducted the experiments, Analyzed the data, Drafted the manuscript, Edited the manuscript; David M Beal, Formal analysis, Investigation, Methodology, Designed the research, Conducted the experiments, Analyzed the data, Edited the manuscript; Nadejda Koloteva-Levine, Investigation, Methodology, Conducted the experiments, Analyzed the data, Edited the manuscript; Tracey J Purton, Investigation, Methodology, Conducted the experiments, Edited the manuscript; Mick F Tuite, Conceptualization, Formal analysis, Investigation, Methodology, Designed the research, Analyzed the data, Edited the manuscript; Wei-Feng Xue, Conceptualization, Resources, Data curation, Software, Formal analysis, Supervision, Funding acquisition, Validation, Investigation, Methodology, Project administration, Designed the research, Wrote the analytical software tools, Analyzed the data, Managed the research, Drafted the manuscript

## Author ORCIDs

Ricardo Marchante (iD) http://orcid.org/0000-0003-3153-6329
Mick F Tuite (iD) https://orcid.org/0000-0002-5214-540X
Wei-Feng Xue (iD) http://orcid.org/0000-0002-6504-0404

## Decision letter and Author response

Decision letter https://doi.org/10.7554/eLife.27109.015
Author response https://doi.org/10.7554/eLife.27109.016

# Additional files

## Supplementary files

• Transparent reporting form
DOI: https://doi.org/10.7554/eLife.27109.014

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
