## [Decision Letter]

Thank you for submitting your article "The physical dimensions of amyloid aggregates control their infective potential as prion particles" for consideration by *eLife*. Your article has been reviewed by three peer reviewers, and the evaluation has been overseen by a Reviewing Editor and Richard Aldrich as the Senior Editor. The following individuals involved in review of your submission have agreed to reveal their identity: Alexander K Buell (Reviewer #1); Margaret Sunde (Reviewer #2); Frederic Rousseau (Reviewer #3).

The reviewers found the work to be of very high quality and of interest and importance. "The study is technically extremely well performed. The data on the fibril length distributions is of exceptionally high quality and the subsequent analysis in terms of aggregation behaviour and in particular infectivity behaviour is highly convincing. " and "This approach developed by the authors is extremely helpful to the community as it allows to design controlled and quantifiable seeding/infectivity experiments to compare different amyloid models"

The reviewers have discussed the reviews with one another and the Reviewing Editor has drafted this decision to help you prepare a revised submission.

Summary:

Marchante et al. quantify the dependence of yeast prion (sup35) infectivity on seed particle size. They find that infectivity requires seed sizes of less than 200nm. They suggest that this that cutoff results from the inability of cells to internalize large particles.

Essential points to address:

1) There is limited correlation between the in vitro studies presented here and infectious Sup35 material. How were the "in vitro formed prion particles" from the [PSI+] extract formed? (from subsection “Characterization of Sup35NM prion particles”, paragraph two)

2) What is the infectivity of the cell extract that is analysed in parallel in Figure 3.e. the control?

3) The effect of the length of the particles on seeding efficiency does not appear to have been tested directly. What is the effect of using equal numbers of long and short particles, to directly test the efficiency of seeding with long particles?

4) Seeding efficiency in vitro correlates with particle concentration (derived from size distribution) in such a manner that the linear fit intersects at (0,0). in vivo PSI+ induction is offset by a 200nM particle size cut-off. The authors attribute this cut-off to the inability of larger particles to enter the cell but they do not show this directly. It would be good that uptake would be directly measured, at least for two extreme conditions (e.g. 30sec vs 960s) to make this point entirely. Please at least comment on this point.

Comments from reviewers you might find helpful in your revision/further work

1) As far as I understand, in the case of yeast prions, amyloid fibrils do not have to enter the cell from the outside, but they are rather transmitted from other to daughter cells during cell division. Also, to my knowledge, the mammalian prion protein is not intracellular. However, things are quite different with for example α-synuclein and tau aggregates, which are intracellular, and which might be able to be transmitted from cell to cell. Therefore, while the question addressed in this work is extremely interesting and relevant, the molecular system with which this has been studied might be less so. However, I see the point of course that this is in some ways the ideal system to test these kinds of things, but only because of the possibility to have a simple readout for infection, not because the question is particularly relevant in this system. This aspect should be commented on by the authors.

2) Another point is the question whether the increased infectivity of short fibrils is simply due to enhanced diffusion. This has been brought forward as one of the explanations why small aggregate species (oligomers) are more toxic to cells – maybe simply because they are more mobile. It would be great if the authors could comment on that, and maybe even plot the fibrillar diffusion coefficients as a function of length (see work by de la Torre for the diffusion coefficients of rods), to demonstrate that there is no dramatic cut off in diffusion coefficient at 200 nm length. This would give additional support for their hypothesis. Looking into diffusion is important in this context also because what might really matter for the transmission of yeast prions in vivo, is how they are diffusing around the cell and into the daughter cells.

3) I am not requesting this as additional experiments (maybe a future study?), but is it possible to check how likely it is for cells to transmit prions to daughter cells, as a function of what length distribution they have been infected with?

4) In amyloid systems (e.g. asyn) for which cell-cell transmission across the membrane could be quite relevant, it appears more and more likely, that it is not fragmentation, but surface catalysis (secondary nucleation) that is responsible for fibril amplification (see e.g. Buell et al., 2014). How does this influence the conclusions of this manuscript? In particular this might lead to the strategy that the authors suggest (make fibrils longer to stop transmission) not working, as longer fibrils can still act as a sites for the formation of secondary nuclei/toxic oligomers, and by making them longer, one produces more surface for that. This should be commented on as well.

5) The size of the infectious Sup35 particles from the cell extract should be determined using sucrose density gradients. While it might be difficult to achieve, could the anti-Sup35 antibody be used to purify the material so that it could be imaged by AFM?

6) The authors have developed methods that should allow them to test their hypothesis with purified preparations from sucrose density gradients that contain only particles of below or above 200 nm. This should be included in the study in order to support their major conclusions and to provide a biological correlate to the in vitro studies.

7) There is evidence that the activity of Hsp104 and other co-chaperones is responsible for fragmentation of Sup35 particles in vivo. Is Hsp104 active on these in vitro preparations? It would be interesting to determine whether activity of Hsp104 results in a population of fibrils shorter than ~200 nm.

---

## [Author Response]

*Essential points to address:*

*1) There is limited correlation between the* in vitro *studies presented here and infectious Sup35 material. How were the "*in vitro *formed prion particles" from the [PSI+] extract formed? (from subsection “Characterization of Sup35NM prion particles”, paragraph two)*

The sentence the referees refer to contains a typographical error, where it reads ‘in vitro’ it should read ‘in vivo’. We thank the referees for pointing out this mistake and we have corrected the text accordingly. We agree with the reviewers that the link between our in vitro generated prion particles and in vivo Sup35 prions seems limited and we apologise for not making this link clearer in our manuscript. Our study was designed to isolate the effect of physical shape and size of the amyloid particles on prion transfection efficiency, and as we show in our paper, we were only able to probe these effects effectively using the synthetic yeast prions as a tractable biophysical tool. In this case, our conclusions are also entirely consistent with size-dependent prion propagation of the prion particles seen in vivo(Derdowski et al., 2010). To further address this point, we have also added new experimental data showing the transfection efficiency of in vivo formed prion particles in the form of a [*PSI*+] cell-free extract (subsection “Characterization of Sup35NM prion particles”, Figure 3—figure supplement 1), added information on the preparation of the cell-free extract in the Material and Methods section and added further clarifications on our approach in isolate the effect of physical shape and size of the amyloid particles using pure in vitroderived material compared to in vivoformed particles in cell-free extracts.

*2) What is the infectivity of the cell extract that is analysed in parallel in Figure 3.e. the control?*

We have now added new experimental data on the transfection efficiency of the [*PSI+*] cell-free extract. (Figure 3—figure supplement 1). As stated in #1 above, we agree with the reviewers that the in vivo-formed prion particles in the cell-free extract and the in vitro generated synthetic prion particles are unlikely to be identical. Here, the Sup35 concentration in the cell-free extract varies depending on the specific expression levels at the point of sample preparation and cell-free extracts will also contain chaperones and other yeast prions that may interfere with the interpretation of the transfection results. Hence, any transfection efficiency difference seen with the cell-free extract cannot be directly attributed to differences in size, shape and suprastrucutre of the amyloid particles in the same way as our synthetic and pure in vitro-generated amyloid samples. We have added this information to the Results section of our manuscript (subsection “Characterization of Sup35NM prion particles”).

*3) The effect of the length of the particles on seeding efficiency does not appear to have been tested directly. What is the effect of using equal numbers of long and short particles, to directly test the efficiency of seeding with long particles?*

The reviewers suggest an important additional experiment that we have now carried out in order to directly test the predictions of our model for seeding efficiency (subsection “Influence of fibril particle concentration and size on prion transfection efficiency”, yellow data points in Figure 5—figure supplement 2, together with recalculated particles concentrations to reflect the final particles concentrations in Figure 5—figure supplement 2). We have also included the results of a new experiment to test the predictions for transfection efficiency (Figure 5—figure supplement 3) in the revised manuscript. In short, our new experimental data show that equal number of particles in fibrils samples with different length distributions indeed confer the same seeding efficiency as predicted. Furthermore, our new transfection experiments also show that equal number of active particles for transfection in fibrils samples with different length distributions do confer the same transfection efficiency as predicted by our particle number and activity, based on our size cut off model. We believe these results significantly strengthen our conclusions and we thank again the referees for this very constructive suggestion.

*4) Seeding efficiency* in vitro *correlates with particle concentration (derived from size distribution) in such a manner that the linear fit intersects at (0,0).* in vivo *PSI+ induction is offset by a 200nM particle size cut-off. The authors attribute this cut-off to the inability of larger particles to enter the cell but they do not show this directly. It would be good that uptake would be directly measured, at least for two extreme conditions (e.g. 30sec vs 960s) to make this point entirely. Please at least comment on this point.*

We very much agree with the reviewer’s comments on this point. In fact, direct observation of a prion particle entering a cell is something that we have tried very hard to do. For example, we have created a Sup35NM variant containing one single cysteine at the Cterminus and labelled it with FITC. We then showed that the labelled fibrils did assemble into amyloid fibrils either on its own or when mixed with wild type Sup35NM. Subsequently, we performed transfection experiments and attempted to detect the fluorescently labelled particles using confocal fluorescence microscopy. Overall, we found that 1) the absolute number of cells taking up prion particles is very low compared to the total original number of cells, 2) the nascently-transfected yeast sphaeroplasts are not stable and physically challenging to handle for confocal microscopy, 3) there is a large dilution factor when single particles enter the cell volume, and 4) the resolution of fluorescence microscopy is not high enough to distinguish the exact localisation of fluorescence signals from single particles. To our knowledge, no study to date has directly detected single amyloid particle uptake in yeast cells. However, this is of course something we are very much interested in continuing to pursue.

Consequently, we have modified the Discussion section to comment on the limitations of our indirect approach without the direct observation of the transfecting particles (paragraph six). In addition, our new data that show transfection efficiency of samples for the two extreme conditions (after 15 s and 960 s of sonication) adjusted to equal particle concentration matched predictions based on our size cut off model, taking into account the predicted activity of the samples at different length (subsection “Influence of fibril particle concentration and size on prion transfection efficiency”, Figure 5—figure supplement 3). We believe that these confirmed predictions considerably strengthen our model.

*Comments from reviewers you might find helpful in your revision/further work*

*1) As far as I understand, in the case of yeast prions, amyloid fibrils do not have to enter the cell from the outside, but they are rather transmitted from other to daughter cells during cell division. Also, to my knowledge, the mammalian prion protein is not intracellular. However, things are quite different with for example α-synuclein and tau aggregates, which are intracellular, and which might be able to be transmitted from cell to cell. Therefore, while the question addressed in this work is extremely interesting and relevant, the molecular system with which this has been studied might be less so. However, I see the point of course that this is in some ways the ideal system to test these kinds of things, but only because of the possibility to have a simple readout for infection, not because the question is particularly relevant in this system. This aspect should be commented on by the authors.*

The reviewer is absolutely correct in that we took the advantage of the well-defined readout our system offered to investigate the effect of physical properties on prion infectivity, with all the benefits and caveats purified biophysical experiments offer. Our results are nevertheless consistent with intracellular prion propagation in yeast as a ‘size-based’ process (Derdowski et al., 2010). We agree with the reviewers that for amyloid systems such as α-synuclein and tau aggregates, intracellular factors as well as the size cut-off we have seen in our study will play important roles. Consequently, we have added discussions on the intracellular and other factors that could further contribute to size and shape effects of prion transmission (Discussion paragraph six).

*2) Another point is the question whether the increased infectivity of short fibrils is simply due to enhanced diffusion. This has been brought forward as one of the explanations why small aggregate species (oligomers) are more toxic to cells – maybe simply because they are more mobile. It would be great if the authors could comment on that, and maybe even plot the fibrillar diffusion coefficients as a function of length (see work by de la Torre for the diffusion coefficients of rods), to demonstrate that there is no dramatic cut off in diffusion coefficient at 200 nm length. This would give additional support for their hypothesis. Looking into diffusion is important in this context also because what might really matter for the transmission of yeast prions* in vivo*, is how they are diffusing around the cell and into the daughter cells.*

We agree with the reviewers that diffusion will be an additional factor. Using the model and the parameters reported by de la Torre and co-workers and the fact that translational diffusion scales with 1/length in this model, diffusion is likely to be a factor for particles smaller than ~50 nm. We have modified the Discussion (paragraph five and Ortega and de la Torre, 2003) to take into account this consideration.

*3) I am not requesting this as additional experiments (maybe a future study?), but is it possible to check how likely it is for cells to transmit prions to daughter cells, as a function of what length distribution they have been infected with?*

This would be extremely challenging to do. As discussed above, we have tried to detect single amyloid particle uptake in yeast cells using labelled Sup35NM without success for various technical reasons. Derdowski et al., 2010) have certainly shown using Sup35-GFP fusions that there is a molecular mass threshold that limits transmission of [*PSI*+] prion aggregates, i.e. propagons, during cell division although length distribution was not considered by these authors. In addition, any in vitro-generated prion particle that enters the cell will undoubtedly undergo change in length distribution and will associate with various endogenous proteins that may modulate transmissibility. Overall, we do agree that direct dynamic detection of prion particle length/size distributions in the cell would be extremely interesting and we are very much interested in continuing to pursue this avenue of research.

*4) In amyloid systems (e.g. asyn) for which cell-cell transmission across the membrane could be quite relevant, it appears more and more likely, that it is not fragmentation, but surface catalysis (secondary nucleation) that is responsible for fibril amplification (see e.g. Buell et al., 2014). How does this influence the conclusions of this manuscript? In particular this might lead to the strategy that the authors suggest (make fibrils longer to stop transmission) not working, as longer fibrils can still act as a sites for the formation of secondary nuclei/toxic oligomers, and by making them longer, one produces more surface for that. This should be commented on as well.*

We agree with the referee that the same active surfaces that promote secondary nucleation may also affect their infective potential, e.g. by altering their fibril-fibril interactions or interactions with other cellular/membrane surfaces. We have modified the Discussion to take these possibilities into account (paragraph five). We have also added that promoting the formation of large inert aggregates of transmissible particles may be a possible therapeutic strategy to reduce the infective potential of the particles (Results section paragraph one). This approach is consistent with both reducing secondary nucleation and reducing their infective activities.

*5) The size of the infectious Sup35 particles from the cell extract should be determined using sucrose density gradients. While it might be difficult to achieve, could the anti-Sup35 antibody be used to purify the material so that it could be imaged by AFM?*

We have indeed tried to image the sucrose density gradients fractionated prion particles, both in vivo and in vitro in origin, using AFM. However, as the reviewers also eluded to in their comment, we found that the high concentration of sucrose used in the gradients unfortunately makes it technically not feasible to image such samples as the sucrose heavily coats the particles and the surface substrate.

*6) The authors have developed methods that should allow them to test their hypothesis with purified preparations from sucrose density gradients that contain only particles of below or above 200 nm. This should be included in the study in order to support their major conclusions and to provide a biological correlate to the* in vitro *studies.*

We agree with the reviewer that the approach we developed in this manuscript open up a range of possible avenues of experiments to test the infective potential of a variety of in vitroand in vivopreparations. As mentioned above, the Sup35 concentration in the cell-free extract varies depending on the specific expression levels in the cell at the time of sample preparation and cell-free extracts may also contain chaperones and other yeast prions that will interfere with the interpretation of the transfection results. In addition, the absolute size of the particles in cell-free extracts cannot be determined by nanoscale imaging as mentioned above. Thus, the effect of size, shape and suprastructure cannot be isolated for a complex mixture such as sucrose gradient fractionated cell-free extracts. Nevertheless, this is an area that we are currently actively investigating.

*7) There is evidence that the activity of Hsp104 and other co-chaperones is responsible for fragmentation of Sup35 particles* in vivo*. Is Hsp104 active on these* in vitro *preparations? It would be interesting to determine whether activity of Hsp104 results in a population of fibrils shorter than ~200 nm.*

We agree with the referees that the molecular mechanism of chaperone-catalysed fibril fragmentation reactions is a very interesting as well as important question to address. While not directly related to this manuscript, this is also something we are currently actively investigating.